# Sustainable Mulberry (*Morus nigra* L., *Morus alba* L. and *Morus rubra* L.) Production in Eastern Turkey

Ahmet Can [1], Ahmet Kazankaya [2], Erdal Orman [3], Muttalip Gundogdu [4], Sezai Ercisli [5], Ravish Choudhary [6] and Rohini Karunakaran [7,*]

1    The Ministry of National Defense, Ankara 06000, Turkey; ahmet.can.tr@gmail.com
2    Department of Horticulture, Faculty of Agriculture, Ahi Evran University, Kırşehir 40100, Turkey; a.kazankaya@ahievran.edu.tr
3    Atatürk Horticultural Central Research Institute, Yalova 77100, Turkey; e.orman77@gmail.com
4    Department of Horticulture, Faculty of Agriculture, Bolu Abant Izzet Baysal University, Bolu 14030, Turkey; gundogdumuttalip@gmail.com
5    Department of Horticulture, Faculty of Agriculture, Atatürk University, Erzurum 25000, Turkey; sercisli@gmail.com
6    Division of Seed Science and Technology, ICAR-Indian Agricultural Research Institute, New Delhi 110012, India; ravianu1110@gmail.com
7    Unit of Biochemistry, Faculty of Medicine, AIMST University, Semeling, Bedong 08100, Malaysia
*    Correspondence: rohini@aimst.edu.my

**Abstract:** In this study, a total of 55 wild-grown mulberry landraces belonging to *Morus alba* L., *Morus rubra* L., and *Morus nigra* L. species (*Rosales* order, *Moraceae* family, *Morus* L. genus) were sampled around the Van Lake basin, and some fruit characteristics were determined. All landraces are naturally grown in the Lake Van basin under pesticide-free conditions. As fruit character, phenolic compounds (gallic acid, catechin, quercetin, protocatechuic acid, vanillic acid, rutin, chlorogenic acid, caffeic acid, syringic acid, *p*-coumaric acid, ferulic acid, and phlorizin) and organic acids (malic acid, citric acid, tartaric acid, succinic acid, fumaric acid and ascorbic acid) were determined. Principal component analysis (PCA) was performed to determine the correlation between mulberry species in terms of biochemical compounds. As a result of PCA-biplot analysis, two variations were sufficient to explain the correlation between phenolic compounds and organic acids. This ratio reveals that mulberry species are separated with sharp boundaries in terms of biochemical compounds. Chlorogenic acid and rutin content were high in all mulberry landraces. The highest chlorogenic acid content was detected in landrace 65VN03 belonging to *M. rubra* (3.778 mg/g), 65GV12 belonging to *M. nigra* (3.526 mg/g), 13AD08 belonging to *M. rubra* (2.461 mg/g), and 13AH02 belonging to *M. rubra* (6.246 mg/g) landraces. In terms of organic acid content, malic acid was the dominant organic acid for genotypes. The rich bioactive compounds make *M. alba*, *M. rubra*, and *M. nigra* landraces as cultivar candidates for breeding purposes. It is a valuable source of bioactive agents that may have prevented humans from oxidative-stress-related diseases.

**Keywords:** mulberry; genotypes; phenolic compounds; organic acids; vitamin C

## 1. Introduction

Turkey is well-known for its rich flora and less-known plants, including fruits, which are widespread with a broad geographic distribution [1–3].

Mulberries have high adaptability to different climatic and soil conditions. The species is resistant to abiotic stress conditions such as low soil fertility or harsh climate, and it readily colonizes wild, edge habitats, fallow pastures, or wastelands. Due to this feature, this less-known fruit is naturally grown in valleys, as well as high-altitude plateaus, even above 2500 m asl [4,5].

They are grown in temperate and subtropical climate conditions because many species and numerous genotypes and landraces represent the genus. The species are quite different

from each other for most of the plant biological traits. *Morus alba* L. (white mulberry), *Morus nigra* L. (black mulberry), and *Morus rubra* L. (red mulberry) species are the most widespread species in the world [6–10]. In Turkey, where temperate, subtropical, and even small tropical climate areas are experienced, many fruit species can be grown as wild for centuries, showing rich biological diversity, and revealing valuable gene resources.

Biodiversity is a scorching topic for humankind, and everyone depends on the world's biological resources for survival. It is very important for sustainable fruit production as well. Developing or developed nations are impoverished by the continuing loss and degradation of agrobiodiversity [11,12]. Their diversity offers the possibility of increasing food supplies and adapting to changing conditions.

Anatolia, which has a long history of using mulberries as edible fruits and traditional medicine, is rich in white, red, and black mulberry genetic resources and exhibits great agrobiodiversity for fruits. In Turkey, mulberry fruit is commonly eaten fresh, dried, or processed into molasses and jam for its delicious taste, pleasing color, low-calorie content, and high nutrient content. For centuries mulberry fruit is also used in folk medicine for its pharmacological effects, including fever reduction, treatment of sore throat, liver and kidney protection, eyesight improvement, and ability to lower blood pressure in Turkey [13–15]. The rich gen pool of mulberries in Turkey is still important due to considerable past and present contributions of farmers and rural communities, especially in the developing world, for the creation, conservation, and availability of wild grown mulberry genetic resources [7–10,13]. In Turkey all mulberry trees are pesticide free due to its strong defense mechanism to pest and diseases. Mulberries have high content of phenolic compounds and it is well-known that phenolic compounds are the key elements of the defense system of plants against pest and diseases.

More recently, there has been an increased interest in the biological traits of mulberry fruits worldwide. All these studies show the characteristic rich components of mulberry fruit and their health benefits [16–19]. Parallel to these studies, the number of studies on mulberry fruit biological traits in Turkey has increased but not enough. The country has diverse climatic and ecological conditions; thus, each region has its specific mulberry genotypes because Turkey has no registered cultivars for mulberries. Therefore, each region needs to work on its mulberry germplasm.

Compared to studies of Asian countries where mulberries originated and are widely cultivated, studies focused on the leaves of mulberries, which are vital for sericulture; however, no commercial sericulture activities are available in Turkey, and mulberries are only used for fruit production [13,20–23].

Black and white mulberries are very popular and have more commercial value than red mulberries. Black mulberry fruits are valued as the most prominent attractive fruits for fresh consumption, have higher phytochemical content, are very suitable for fruit juice production, and are the most used species in folk medicine for a long time compared to white and other mulberries. However, white mulberries widely process into several valuable products such as molasses and constituents of both species are suggested as promising natural sources for dietary supplements and cosmetic products [24–29].

There are profound biological differences between both the species. Black mulberry is very difficult to propagate and grows very slowly. In addition, black mulberry is more resistant to adverse environmental conditions than white mulberry. The high phenolic compound content in various organs of the black mulberry plant constitutes the plant's defense system [29,30].

This study determines the phenolic compounds and organic acids of landraces of mulberry species (*Morus nigra* L., *Morus alba* L., and *Morus rubra* L.) grown around Lake Van. Many black, white, and red mulberry species are grown in the natural flora in this region where the research was carried out. However, since this affluent mulberry population is not sufficiently utilized, it has been seen that mulberry genetic resources are in danger of extinction day by day. Therefore, disseminating research on these genotypes is of great importance in protecting mulberry genetic resources and world biodiversity.

## 2. Materials and Methods

### 2.1. Plant Materials

In this study, phenolic compounds and organic acids of 55 seed-propagated mulberry landraces grown in Van Merkez, and Gevaş districts in the Van Lake basin, and Adilcevaz and Ahlat districts in Bitlis province [31] were determined (Figures 1 and 2). Traditionally, Turkish farmers left good seed-propagated mulberry plants on the field. In this way, 55 landraces are so-called farmers' selection. No pesticides and inorganic fertilizers were used in these landraces. The landraces selected in this study are those grown in natural form. The surface area of Lake Van is 3713 km$^2$. Lake Van is the world's largest soda lake and also the largest lake in Turkey. Van climate is continental climate and vegetation is steppe. Winter months are harsh in Van with heavy snow. Spring months are the months with the most precipitation in Van. Summer months in Van are generally dry and hot, with the effect of Lake Van making it somewhat cooler. These months do not have much precipitation, but are accompanied by winds. The annual average temperature is 9 °C. The average temperature of the coldest month of the year is −3.5 °C, and the average temperature of the warmest month is 22 °C [32]. Microclimate areas have been formed in the region due to the moderating effect of Van Lake on the climate. It was observed that mulberry species spread in the natural flora, especially in the Van Lake basin and form an affluent population. This situation allows the cultivation of many fruit species in the region. In June-July, the ripening period of the mulberry fruits, 1 kg of fruit samples was taken from each tree in a homogeneous way representing the genotypes and placed in cloth bags. In this study, fruits belonging to mulberry species were harvested at full maturity stages with species-specific color formation. These landraces, which grow in a wide area in nature, are grown in areas with different soil structures. It was observed that landraces of mulberry species were generally between the ages of 20 and 25 in the region where the study was conducted. Landraces of *Morus alba* L. species were harvested on 25–30 June 2015–2016, genotypes of *Morus rubra* L. species were harvested on 13–18 July 2015–2016, and landraces of *Morus nigra* L. were harvested on 7–19 July 2015–2016 in Merkez and Gevaş districts located in Van province. Harvest dates of mulberry species in the Bitlis region; landraces of *Morus alba* L. species in Adilcevaz and Ahlat districts were harvested on 20–28 July 2015–2016, *Morus rubra* L. species were harvested on 3–12 July 2015–2016, and *Morus nigra* L. landraces were harvested on 11–16 July 2015–2016. Then the necessary analyses of these samples were made in the laboratory. Samples were stored at −80 °C until analysis. Biochemical analyses were done in triplicate.

### 2.2. Determination of Phenolic Compounds

Phenolic compounds were detected with a modified HPLC procedure suggested by Rodriguez-Delgado et al. [33]. Fruit extracts were mixed with distilled water in a ratio of 1:1. The mixture was centrifuged for 15 min at 15,000 rpm. Supernatants were filtrated with coarse filter paper and twice with 0.45-μm membrane filter (Millipore Millex-HV Hydrophilic PVDF, Millipore, Burlington, MA, USA) and injected into an HPLC (Agilent, Santa Clara, CA, USA). Chromatographic separation was performed with a 250 × 4.6 mm, 4 μm ODS column (HiChrom, Reading, UK). As mobile phase, solvent A methanol: acetic acid: water (10:2:28) and Solvent B methanol: acetic acid: water (90:2:8) were used. Spectral measurements were made at 254 and 280 nm, and flow rate and injection volume were adjusted to 1 mL min$^{-1}$ and 20 μL, respectively.

### 2.3. Determination of Vitamin C

Vitamin C content of fruits was detected with a modified HPLC procedure suggested by [34]. Fruit extracts (50 g) were supplemented with 2.5% ($w\,v^{-1}$) metaphosphoric acid (Sigma, M6285, 33.5%), then centrifuged at 6500 rpm for 10 min at 4 °C temperature. A 0.5 mL aliquot of the mixture was brought to a final volume of 10 mL with 2.5% ($w\,v^{-1}$) metaphosphoric acid. Supernatants were filtered with a 0.45 μm PTFE syringe filter (Phenomenex, Reading, UK). C18 column (Phenomenex Luna C18, 250 × 4.60 mm, five μ)

was used to identify ascorbic acid at 25 °C. Double distilled water with 1 mL min$^{-1}$ flow rate and pH of 2.2 (acidified with $H_2SO_4$) was used as a mobile phase. Spectral measurements were made at 254 nm wavelength using a DAD detector. Different standards of Vitamin C (Sigma, A5960) (50 mg kg$^{-1}$, 100 mg kg$^{-1}$, 500 mg kg$^{-1}$, 1000 mg kg$^{-1}$ and 2000 mg kg$^{-1}$) were used for quantification.

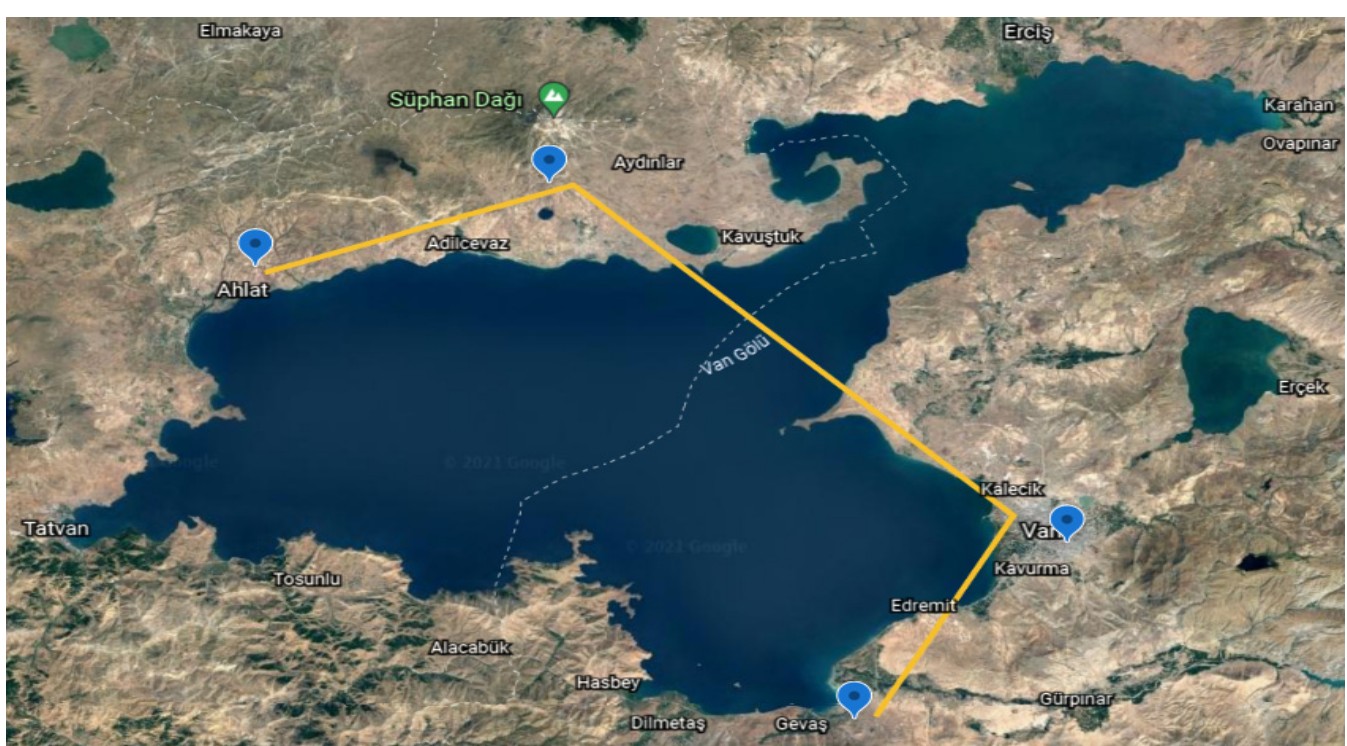

**Figure 1.** Distribution areas of mulberry genotypes around Van Lake.

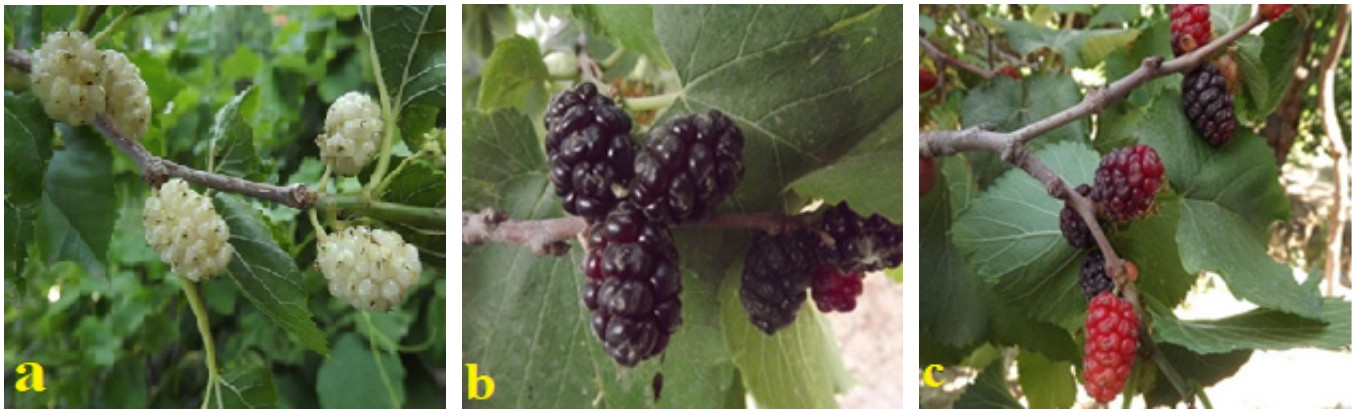

**Figure 2.** (**a**): Morus alba fruits, (**b**): Morus nigra fruits, (**c**): Morus rubra fruits.

### 2.4. Determination of Organic Acids

Organic acids were identified by the technique reported by [35]. Each sample (50 g) was mixed with 80 mL of 0.009 N $H_2SO_4$ (Heidolph Silent Crusher M, Schwabach, Germany), then homogenized for 1 h with a shaker (HeidolphUnimax 1010, Germany). The mixture was centrifuged for 15 min at 15,000 rpm, and supernatants were filtrated twice with 0.45 µm membrane filter following filtration with coarse filter (Millipore Millex-HV Hydrophilic PVDF, Millipore, Burlingron, MA, USA) and run through a SEP-PAK C18 cartridge. Organic acid readings were performed with HPLC using Aminex column (HPX-

87 H, 300 mm × 7.8 mm, Bio-Rad Laboratories, Richmond, CA, USA) at 214 and 280 nm wavelengths Agilent package program (Agilent, Santa Clara, CA, USA).

### 2.5. Statistical Analysis

The Duncan test was used as a multiple comparison test to express the differences between the averages. Windows SPSS 20 was used in the statistical evaluations, and the differences between the means were evaluated by subjecting to ANOVA variance analysis and determined with Duncan multiple comparison tests ($p < 0.05$). In R software, the principal component analysis was used for all variables with the ggplot2 and factoextra packages [36].

## 3. Results

### 3.1. Organic Acids

In this study, 55 landraces of 3 mulberry species growing in the natural flora of the Van Lake basin were examined, and the agro-biological characteristics in fruits of these landraces were determined. Among organic acids, malic acid was higher than other organic acids and dominant organic acids for mulberry fruits. Differences between mulberry landraces in terms of organic acid content were statistically significant ($p \leq 0.05$).

When the malic acid levels of genotypes in Van Merkez were examined, the highest value was found in 65VN09, *M. alba* (5.725 mg 100g$^{-1}$) landrace, and the lowest value was determined in 65VN04, *M. rubra* (1.923 mg 100 g$^{-1}$) (Table 1). The malic acid content of mulberry landraces grown in the Gevaş (Van) district varied between 2.981 and 7.918 g/100 g (Table 2), followed by citric acid (0.770–2.710 g/100 g).

**Table 1.** Organic acid (mg 100 g$^{-1}$) content of mulberry landraces in the central district of Van.

| Species | Landraces | Malic Acid | Citric Acid | Succinic Acid | Tartaric Acid | Fumaric Acid | Vitamin C |
|---|---|---|---|---|---|---|---|
| *M. rubra* | 65VN01 | 2.585 ± 0.003 [f,*] | 2.238 ± 0.002 [d] | 0.887 ± 0.005 [f] | 0.101 ± 0.002 [i] | 0.083 ± 0.003 [f] | 20.179 ± 0.004 [a] |
| | 65VN02 | 3.356 ± 0.003 [d] | 1.941 ± 0.009 [e] | 1.009 ± 0.002 [e] | 0.159 ± 0.001 [e] | 0.109 ± 0.001 [d] | 15.357 ± 0.010 [c] |
| | 65VN03 | 2.967 ± 0.001 [e] | 1.894 ± 0.002 [f] | 1.014 ± 0.005 [de] | 0.197 ± 0.001 [b] | 0.098 ± 0.002 [e] | 15.276 ± 0.005 [c] |
| | 65VN04 | 1.973 ± 0.003 [j] | 0.771 ± 0.007 [i] | 0.654 ± 0.004 [i] | 0.181 ± 0.004 [c] | 0.048 ± 0.002 [i] | 11.465 ± 0.007 [f] |
| | 65VN05 | 2.304 ± 0.004 [g] | 1.719 ± 0.006 [h] | 0.666 ± 0.005 [h] | 0.124 ± 0.002 [g] | 0.145 ± 0.005 [b] | 10.009 ± 0.007 [g] |
| *M. alba* | 65VN06 | 2.104 ± 0.002 [i] | 1.868 ± 0.005 [g] | 0.838 ± 0.002 [g] | 0.189 ± 0.003 [d] | 0.096 ± 0.002 [e] | 18.055 ± 0.007 [b] |
| | 65VN07 | 4.074 ± 0.009 [c] | 3.937 ± 0.004 [a] | 1.667 ± 0.004 [a] | 0.210 ± 0.002 [a] | 0.117 ± 0.003 [c] | 13.015 ± 0.013 [e] |
| | 65VN08 | 4.578 ± 0.005 [b] | 3.139 ± 0.008 [c] | 1.036 ± 0.003 [c] | 0.141 ± 0.001 [f] | 0.075 ± 0.002 [g] | 14.042 ± 0.015 [d] |
| | 65VN09 | 5.725 ± 0.003 [a] | 3.621 ± 0.002 [b] | 1.018 ± 0.003 [d] | 0.106 ± 0.002 [h] | 0.433 ± 0.002 [a] | 9.050 ± 0.007 [h] |
| *M. nigra* | 65VN10 | 2.233 ± 0.006 [h] | 0.717 ± 0.002 [j] | 1.127 ± 0.006 [b] | 0.126 ± 0.002 [g] | 0.057 ± 0.002 [h] | 12.552 ± 0.490 [e] |

\* The difference among the means indicated with the same lower-case letter in columns was not significant ($p \leq 0.05$).

**Table 2.** Organic acid (mg 100 g$^{-1}$) content of mulberry landraces in the Gevaş District of Van.

| Species | Landraces | Malic Acid | Citric Acid | Succinic Acid | Tartaric Acid | Fumaric Acid | Vitamin C |
|---|---|---|---|---|---|---|---|
| *M. rubra* | 65GV01 | 7.576 ± 0.008 [b,*] | 2.710 ± 0.010 [a] | 0.208 ± 0.003 [h] | 0.132 ± 0.004 [g] | 0.115 ± 0.002 [i] | 15.018 ± 0.014 [e] |
| | 65GV02 | 7.918 ± 0.012 [a] | 1.605 ± 0.003 [d] | 0.216 ± 0.003 [g] | 0.112 ± 0.004 [h] | 0.274 ± 0.004 [e] | 14.285 ± 0.011 [g] |
| | 65GV03 | 6.848 ± 0.006 [f] | 2.543 ± 0.004 [b] | 0.106 ± 0.001 [l] | 0.231 ± 0.003 [d] | 0.482 ± 0.007 [a] | 15.157 ± 0.012 [d] |
| | 65GV04 | 5.753 ± 0.005 [g] | 1.017 ± 0.008 [e] | 0.153 ± 0.002 [i] | 0.170 ± 0.002 [e] | 0.333 ± 0.004 [d] | 13.269 ± 0.010 [h] |
| *M. alba* | 65GV05 | 4.559 ± 0.007 [h] | 2.118 ± 0.006 [c] | 0.349 ± 0.004 [d] | 0.169 ± 0.005 [e] | 0.140 ± 0.004 [h] | 9.009 ± 0.008 [k] |
| | 65GV06 | 3.157 ± 0.006 [i] | 2.495 ± 0.004 [b] | 0.758 ± 0.006 [a] | 0.440 ± 0.006 [b] | 0.409 ± 0.021 [b] | 11.553 ± 0.011 [i] |
| | 65GV07 | 3.093 ± 0.005 [k] | 2.458 ± 0.211 [b] | 0.322 ± 0.002 [f] | 0.291 ± 0.001 [c] | 0.134 ± 0.005 [h] | 11.019 ± 0.011 [j] |
| | 65GV08 | 3.106 ± 0.005 [j] | 2.657 ± 0.004 [a] | 0.341 ± 0.003 [e] | 0.435 ± 0.005 [b] | 0.185 ± 0.004 [f] | 16.882 ± 0.010 [b] |
| | 65GV09 | 2.981 ± 0.005 [l] | 2.523 ± 0.002 [b] | 0.393 ± 0.005 [c] | 0.601 ± 0.003 [a] | 0.391 ± 0.002 [c] | 17.277 ± 0.006 [a] |
| | 65GV10 | 7.206 ± 0.006 [e] | 1.698 ± 0.003 [d] | 0.483 ± 0.003 [b] | 0.103 ± 0.001 [i] | 0.162 ± 0.002 [g] | 15.016 ± 0.009 [f] |
| *M. nigra* | 65GV11 | 7.227 ± 0.010 [d] | 0.770 ± 0.003 [f] | 0.124 ± 0.002 [k] | 0.136 ± 0.003 [f,g] | 0.172 ± 0.005 [g] | 8.886 ± 0.010 [l] |
| | 65GV12 | 7.452 ± 0.006 [c] | 1.011 ± 0.003 [e] | 0.140 ± 0.010 [j] | 0.140 ± 0.003 [f] | 0.083 ± 0.001 [j] | 15.354 ± 0.007 [c] |

\* The difference among the means indicated with the same lower-case letter in columns was not significant ($p \leq 0.05$).

The citric acid, tartaric acid, malic acid, succinic acid, fumaric acid, and ascorbic acid (vitamin C) contents of mulberry landraces grown in Adilcevaz (Bitlis) district were varied between 0.999 mg 100 g$^{-1}$–3.679 mg 100 g$^{-1}$, 0.085 mg 100 g$^{-1}$–0.377 mg 100 g$^{-1}$, 2.222 mg 100 g$^{-1}$–6.230 mg 100 g$^{-1}$, 0.067 mg 100 g$^{-1}$–0.461 mg 100 g$^{-1}$, 0.016 mg 100 g$^{-1}$– 0.439 mg 100 g$^{-1}$, 9.566 mg 100 g$^{-1}$–24.077 mg 100 g$^{-1}$, respectively (Table 3). It was observed that the order of organic acid contents of mulberry landraces grown in the Ahlat

(Bitlis) district from highest to lowest was generally in the form of malic acid, citric acid, tartaric acid, succinic acid, and fumaric acid (Table 4). When the organic acid contents of the fruits belonging to the landraces were examined, it was determined that malic acid was higher than the other acids in general. It was observed that the malic acid content of the fruits belonging to the landraces grown, especially in the Gevaş district, was higher than the landraces grown in other regions. Considering the organic acid distribution, it was determined that tartaric acid and fumaric acid contents were lower than other organic acids. In terms of vitamin C content, it was determined that the 13AD15 *M. alba* landrace was superior to the landraces grown in other regions. Vitamin C varied between 9.050 and 20.179 mg 100 g$^{-1}$ among genotypes based on all 55 landraces.

**Table 3.** Organic acid (mg 100 g$^{-1}$) content of mulberry landraces in the Adilcevaz District of Bitlis.

| Species | Landraces | Malic Acid | Citric Acid | Succinic Acid | Tartaric Acid | Fumaric Acid | Vitamin C |
|---|---|---|---|---|---|---|---|
| | 13AD01 | 3.392 ± 0.003 [h,*] | 1.373 ± 0.006 [n] | 0.148 ± 0.002 [h] | 0.099 ± 0.002 [l] | 0.038 ±0.001 [k] | 21.387 ± 0.009 [b] |
| | 13AD02 | 2.769 ± 0.006 [o] | 2.280 ± 0.006 [h] | 0.264 ± 0.001 [d] | 0.109 ± 0.002 [k] | 0.028 ± 0.000 [m] | 12.179 ± 0.005 [l] |
| | 13AD03 | 3.678 ± 0.007 [g] | 2.009 ± 0.007 [j] | 0.155 ± 0.000 [g] | 0.116 ± 0.001 [j] | 0.025 ± 0.001 [n] | 11.021 ± 0.013 [o] |
| | 13AD04 | 2.755 ± 0.005 [p] | 1.114 ± 0.003 [s] | 0.153 ± 0.004 [g] | 0.123 ± 0.002 [i] | 0.022 ± 0.002 [n] | 13.149 ± 0.004 [j] |
| | 13AD05 | 3.051 ± 0.002 [k] | 1.159 ± 0.002 [p] | 0.074 ± 0.001 [m] | 0.158 ± 0.002 [e] | 0.023 ± 0.000 [n] | 13.127 ± 0.010 [j] |
| *M.rubra* | 13AD06 | 2.886 ± 0.003 [n] | 0.999 ± 0.003 [u] | 0.152 ± 0.002 [g] | 0.130 ± 0.003 [h] | 0.016 ± 0.000 [o] | 9.566 ± 0.014 [s] |
| | 13AD07 | 3.358 ± 0.005 [i] | 2.581 ± 0.003 [f] | 0.101 ± 0.002 [l] | 0.159 ± 0.005 [e] | 0.062 ± 0.000 [h] | 10.017 ± 0.008 [r] |
| | 13AD08 | 3.786 ± 0.004 [e] | 2.366 ± 0.006 [g] | 0.114 ± 0.004 [j] | 0.171 ± 0.002 [c] | 0.164 ± 0.004 [e] | 10.884 ± 0.008 [p] |
| | 13AD09 | 2.443 ± 0.003 [r] | 1.144 ± 0.005 [r] | 0.106 ± 0.001 [k] | 0.200 ± 0.002 [b] | 0.323 ± 0.003 [b] | 18.084 ± 0.084 [f] |
| | 13AD10 | 3.762 ± 0.007 [f] | 3.077 ± 0.003 [d] | 0.367 ± 0.003 [b] | 0.377 ± 0.003 [a] | 0.176 ± 0.004 [d] | 13.382 ± 0.008 [i] |
| | 13AD11 | 4.067 ± 0.007 [c] | 3.335 ± 0.005 [b] | 0.174 ± 0.004 [e] | 0.139 ± 0.003 [g] | 0.250 ± 0.001 [c] | 11.016 ± 0.007 [o] |
| | 13AD12 | 2.927 ± 0.008 [m] | 1.528 ± 0.005 [m] | 0.122 ± 0.003 [i] | 0.139 ± 0.001 [g] | 0.051 ± 0.000 [j] | 11.162 ± 0.005 [n] |
| | 13AD13 | 2.222 ± 0.008 [t] | 1.828 ± 0.007 [k] | 0.168 ± 0.001 [f] | 0.113 ± 0.004 [j] | 0.120 ± 0.001 [g] | 21.191 ± 0.009 [c] |
| | 13AD14 | 2.321 ± 0.003 [s] | 2.883 ± 0.002 [e] | 0.113 ± 0.003 [j] | 0.167 ± 0.002 [d] | 0.061 ± 0.001 [hi] | 20.021 ± 0.018 [d] |
| *M. alba* | 13AD15 | 3.919 ± 0.005 [d] | 3.259 ± 0.004 [c] | 0.461 ± 0.005 [a] | 0.199 ± 0.003 [b] | 0.439 ± 0.003 [a] | 24.077 ± 0.008 [a] |
| | 13AD16 | 6.230 ± 0.003 [a] | 3.679 ± 0.005 [a] | 0.287 ± 0.003 [c] | 0.085 ± 0.000 [n] | 0.155 ± 0.003 [f] | 15.078 ± 0.006 [h] |
| | 13AD17 | 2.773 ± 0.006 [o] | 1.722 ± 0.004 [l] | 0.103 ± 0.000 [k,l] | 0.094 ± 0.001 [m] | 0.036 ± 0.000 [k] | 19.085 ± 0.013 [e] |
| *M. nigra* | 13AD18 | 4.233 ± 0.006 [b] | 1.064 ± 0.005 [t] | 0.075 ± 0.002 [m] | 0.199 ± 0.003 [b] | 0.064 ± 0.001 [h] | 16.746 ± 0.008 [g] |
| | 13AD19 | 3.330 ± 0.009 [j] | 2.134 ± 0.005 [i] | 0.067 ± 0.001 [n] | 0.129 ± 0.002 [h] | 0.059 ± 0.001 [i] | 12.021 ± 0.013 [m] |
| | 13AD20 | 2.989 ± 0.006 [l] | 1.320 ± 0.004 [o] | 0.074 ± 0.002 [m] | 0.150 ± 0.003 [f] | 0.032 ± 0.000 [l] | 13.074 ± 0.012 [k] |

* The difference among the means indicated with the same lower-case letter in columns was not significant ($p \leq 0.05$).

**Table 4.** Organic acids (mg 100 g$^{-1}$) content of mulberry landraces in the Ahlat District of Bitlis.

| Species | Landraces | Malic Acid | Citric Acid | Succinic Acid | Tartaric Acid | Fumaric Acid | Vitamin C |
|---|---|---|---|---|---|---|---|
| | 13AH01 | 1.708 ± 0.003 [m,*] | 0.938 ± 0.005 [l] | 0.152 ± 0.005 [i] | 0.105 ± 0.002 [e] | 0.035 ± 0.001 [k] | 12.128 ± 0.003 [f] |
| *M. rubra* | 13AH02 | 6.414 ± 0.010 [c] | 0.855 ± 0.004 [m] | 0.103 ± 0.002 [j] | 0.098 ± 0.001 [f] | 0.019 ± 0.001 [l] | 14.332 ± 0.015 [e] |
| | 13AH03 | 2.006 ± 0.005 [k] | 1.395 ± 0.002 [g] | 0.334 ± 0.005 [d] | 0.313 ± 0.006 [a] | 0.096 ± 0.002 [g] | 10.749 ± 0.006 [g] |
| | 13AH04 | 2.878 ± 0.003 [j] | 2.781 ± 0.005 [f] | 0.344 ± 0.004 [c] | 0.147 ± 0.002 [d] | 0.112 ± 0.002 [d] | 15.156 ± 0.994 [d] |
| | 13AH05 | 5.009 ± 0.008 [g] | 4.987 ± 0.004 [b] | 0.294 ± 0.002 [e] | 0.269 ± 0.007 [b] | 0.091 ± 0.002 [h] | 18.173 ± 0.003 [b] |
| | 13AH06 | 1.984 ± 0.004 [l] | 1.210 ± 0.007 [i] | 0.336 ± 0.005 [d] | 0.314 ± 0.002 [a] | 0.196 ± 0.002 [b] | 16.113 ± 0.006 [c] |
| | 13AH07 | 4.560 ± 0.008 [h] | 1.118 ± 0.002 [j] | 0.546 ± 0.004 [a] | 0.172 ± 0.002 [c] | 0.213 ± 0.001 [a] | 20.213 ± 0.008 [a] |
| | 13AH08 | 3.959 ± 0.003 [i] | 1.011 ±0.003 [k] | 0.284 ± 0.002 [f] | 0.100 ± 0.001 [ef] | 0.103 ± 0.003 [f] | 16.157 ± 0.006 [c] |
| *M. alba* | 13AH09 | 6.024 ± 0.005 [e] | 1.232 ± 0.003 [h] | 0.422 ± 0.003 [b] | 0.313 ± 0.003 [a] | 0.109 ± 0.001 [e] | 10.669 ± 0.007 [g] |
| | 13AH10 | 5.457 ± 0.006 [f] | 3.979 ± 0.001 [e] | 0.203 ± 0.004 [h] | 0.077 ± 0.001 [g] | 0.069 ± 0.001 [i] | 11.140 ± 0.006 [g] |
| | 13AH11 | 7.122 ± 0.005 [b] | 4.322 ± 0.005 [c] | 0.288 ± 0.003 [f] | 0.096 ± 0.001 [f] | 0.098 ± 0.001 [g] | 9.641 ± 0.007 [h] |
| | 13AH12 | 7.228 ± 0.006 [a] | 4.017 ± 0.006 [d] | 0.349 ± 0.002 [c] | 0.073 ± 0.003 [g,h] | 0.175 ± 0.001 [c] | 11.079 ± 0.012 [g] |
| | 13AH13 | 6.323 ± 0.007 [d] | 5.023 ± 0.006 [a] | 0.216 ± 0.004 [g] | 0.069 ± 0.000 [h] | 0.063 ± 0.001 [j] | 10.019 ± 0.018 [h] |

* The difference among the means indicated with the same lower-case letter in columns was not significant ($p \leq 0.05$).

In this study, principal component analysis (PCA) was performed to determine the correlation between mulberry species in terms of organic acid contents. In the PCA analysis, the statistical variation rate was found to be 100%. This ratio was seen as a practical and high value in determining the correlation between mulberry species. It was determined that the fruits of *Morus alba* species contain higher levels of vitamin C than other species. *Morus rubra* was the most prominent species in terms of fumaric, succinic, and tartaric acid content. Morus nigra species was found to be behind other species in terms of organic acid content in general. According to PCA analysis, a negative correlation was determined between malic acid and vitamin C. A parallel correlation was noted between tartaric, succinic, and fumaric acid (Figure 3).

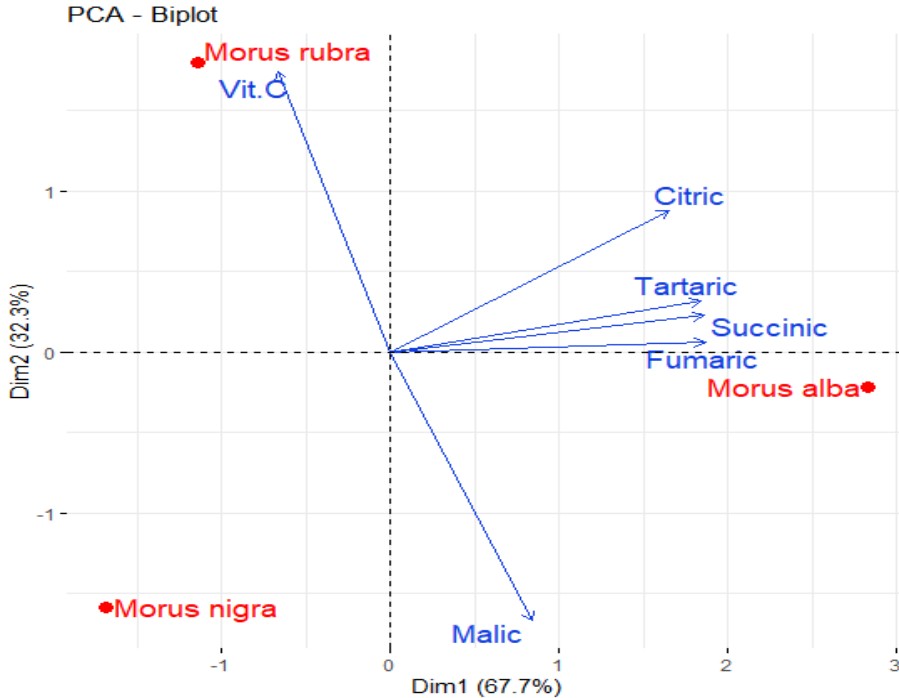

**Figure 3.** Correlation between mulberry species in terms of organic acid content.

### 3.2. Phenolic Compounds

When the phenolic compound contents of the mulberry landraces were examined, it was seen that the dominant phenolic compound was chlorogenic acid, followed by rutin. There were statistically significant differences between mulberry genotypes in terms of phenolic compound contents ($p \leq 0.05$). This study determined that the rutin content of mulberry landraces grown in the central district of Van ranged from 0.115 mg/g (65VN07, *M. alba*) to 2.070 mg/g (65VN03, *M. rubra*).

The chlorogenic acid content ranged from 0.101 mg/g (65VN01, *M. rubra*) to 3.778 mg g$^{-1}$ in 65VN03, *M. rubra* (Tables 5 and 6). The highest chlorogenic acid content of mulberry landraces grown in the Gevaş (Van) district was determined in 65GV12, *M. nigra* (3.526 mg g$^{-1}$) landrace and the highest rutin content was determined in 65GV11, *M. nigra* (3.903 mg g$^{-1}$) landrace (Tables 7 and 8). Among the mulberry landraces grown in the Adilcevaz district of Bitlis province, it was noted that the 13AD08, *M. rubra* genotype was superior to the other landraces in terms of chlorogenic acid content. The highest rutin content was determined in the 13AD02, *M. rubra* landrace (Tables 9 and 10). The highest chlorogenic acid content of mulberry landraces grown in the Ahlat (Bitlis) district was determined in 13AH02, *M. rubra* landrace (6.246 mg g$^{-1}$). The highest rutin content was recorded in the 13AH04, *M. rubra* (2.229 mg g$^{-1}$) landrace (Tables 11 and 12).

As a result of the PCA analysis performed to determine the correlation between mulberry species in terms of phenolic compounds, it was observed that the variation occurred was 100%. This ratio shows that the species are entirely separated from each other, and the correlation between the phenolic compounds is effectively explained by the two PCA factors (Figure 4). The study determined that mulberry species were split into different regions in the PCA plane regarding phenolic compound content. It was observed that Morus nigra species and rutin, chlorogenic, vanillic, and caffeic acid were located in the same region on the PCA plane and were more prominent in terms of these biochemical compounds. Morus alba was found to be superior to other genotypes in terms of gallic, ferulic, and syringic acids. In terms of catechin and protocatechuic acid, it was determined that Morus rubra species came to the fore. It was determined that there

was a negative relationship between catechin and *p*-coumaric and between quercetin and protocatechuic acid.

**Table 5.** Protocatechuic, vanillic, rutin, quercetin, gallic, and catechin content of mulberry landraces in the Central District of Van (mg g$^{-1}$).

| Species | Landraces | Protocatechuic | Vanillic | Rutin | Quercetin | Gallic | Catechin |
|---|---|---|---|---|---|---|---|
| *M. rubra* | 65VN01 | 0.074 ± 0.002 [h,*] | 0.041 ± 0.000 [f] | 0.653 ± 0.002 [d] | 0.152 ± 0.001 [a] | 0.120 ± 0.002 [e] | 0.038 ± 0.000 [e] |
| | 65VN02 | 0.182 ± 0.006 [c] | 0.125 ± 0.001 [d] | 0.694 ± 0.003 [c] | 0.145 ± 0.001 [b] | 0.108 ± 0.003 [f] | 0.319 ± 0.262 [a] |
| | 65VN03 | 0.327 ± 0.002 [a] | 0.350 ± 0.006 [a] | 2.070 ± 0.003 [a] | 0.144 ± 0.001 [b] | 0.097 ± 0.002 [g] | 0.111 ± 0.002 [c] |
| | 65VN04 | 0.233 ± 0.003 [b] | 0.201 ± 0.003 [c] | 1.432 ± 0.006 [b] | 0.155 ± 0.003 [a] | 0.153 ± 0.001 [d] | 0.251 ± 0.001 [b] |
| | 65VN05 | 0.084 ± 0.002 [f] | 0.096 ± 0.004 [e] | 0.128 ± 0.004 [g] | 0.132 ± 0.004 [c] | 0.156 ± 0.001 [c] | 0.056 ± 0.002 [d] |
| *M. alba* | 65VN06 | 0.078 ± 0.001 [g,h] | 0.095 ± 0.002 [e] | 0.155 ± 0.003 [f] | 0.143 ± 0.002 [b] | 0.153 ± 0.003 [d] | 0.021 ± 0.000 [f] |
| | 65VN07 | 0.080 ± 0.001 [f,g] | 0.090 ± 0.003 [e] | 0.115 ± 0.003 [h] | 0.134 ± 0.001 [c] | 0.171 ± 0.001 [a] | 0.017 ± 0.000 [f] |
| | 65VN08 | 0.090 ± 0.001 [e] | 0.090 ± 0.003 [e] | 0.153 ± 0.004 [f] | 0.136 ± 0.001 [c] | 0.165 ± 0.001 [b] | 0.019 ± 0.001 [f] |
| | 65VN09 | 0.103 ± 0.007 [d] | 0.094 ± 0.002 [e] | 0.125 ± 0.001 [g] | 0.135 ± 0.002 [c] | 0.166 ± 0.001 [b] | 0.015 ± 0.000 [f] |
| *M. nigra* | 65VN10 | 0.079 ± 0.003 [f,g,h] | 0.270 ± 0.007 [b] | 0.511 ± 0.006 [e] | 0.133 ± 0.002 [c] | 0.164 ± 0.000 [b] | 0.035 ± 0.002 [e] |

* The difference among the means indicated with the same lower-case letter in columns was not significant ($p \leq 0.05$).

**Table 6.** Chlorogenic, caffeic, syringic, *p*-coumaric, ferulic, and phlorizin content of mulberry landraces in the Central District of Van (mg g$^{-1}$).

| Species | Landraces | Chlorogenic | Caffeic | Syringic | *p*-Coumaric | Ferulic | Phlorizin |
|---|---|---|---|---|---|---|---|
| *M. rubra* | 65VN01 | 0.101 ± 0.002 [e,*] | 0.226 ± 0.002 [d] | 0.140 ± 0.002 [c] | 0.088 ± 0.001 [b,c] | 0.095 ± 0.004 [a] | 0.026 ± 0.001 [b] |
| | 65VN02 | 1.240 ± 0.001 [c] | 0.227 ± 0.003 [c] | 0.145 ± 0.002 [b] | 0.089 ± 0.003 [b] | 0.094 ± 0.001 [a,b] | 0.022 ± 0.001 [c] |
| | 65VN03 | 3.778 ± 0.011 [a] | 0.451 ± 0.006 [a] | 0.122 ± 0.001 [f] | 0.097 ± 0.001 [a] | 0.091 ± 0.001 [a,b] | 0.033 ± 0.000 [a] |
| | 65VN04 | 1.136 ± 0.018 [c] | 0.248 ± 0.002 [c] | 0.120 ± 0.002 [fg] | 0.085 ± 0.001 [c,d] | 0.092 ± 0.003 [a,b] | 0.019 ± 0.000 [d] |
| | 65VN05 | 0.145 ± 0.002 [e] | 0.099 ± 0.004 [f] | 0.118 ± 0.001 [g] | 0.089 ± 0.004 [b] | 0.090 ± 0.001 [b] | 0.011 ± 0.001 [f,g] |
| *M. alba* | 65VN06 | 0.662 ± 0.538 [d] | 0.115 ± 0.004 [e] | 0.123 ± 0.001 [f] | 0.083 ± 0.001 [d] | 0.090 ± 0.002 [b] | 0.011 ± 0.001 [f,g] |
| | 65VN07 | 0.103 ± 0.001 [e] | 0.088 ± 0.003 [g] | 0.135 ± 0.001 [d] | 0.079 ± 0.001 [e,f] | 0.094 ± 0.002 [a,b] | 0.011 ± 0.001 [f,g] |
| | 65VN08 | 0.105 ± 0.002 [e] | 0.098 ± 0.001 [f] | 0.155 ± 0.004 [a] | 0.076 ± 0.000 [f] | 0.095 ± 0.005 [a] | 0.012 ± 0.001 [f,g] |
| | 65VN09 | 0.135 ± 0.002 [e] | 0.101 ± 0.003 [f] | 0.126 ± 0.003 [e] | 0.082 ± 0.001 [d,e] | 0.091 ± 0.003 [a,b] | 0.011 ± 0.001 [f,g] |
| *M. nigra* | 65VN10 | 1.778 ± 0.004 [b] | 0.356 ± 0.003 [b] | 0.122 ± 0.001 [f] | 0.095 ± 0.003 [a] | 0.091 ± 0.002 [a,b] | 0.016 ± 0.000 [e] |

* The difference among the means indicated with the same lower-case letter in columns was not significant ($p \leq 0.05$).

**Table 7.** Protocatechuic, vanillic, rutin, quercetin, gallic and catechin content of mulberry landraces in the Gevaş District of Van (mg g$^{-1}$).

| Species | Landraces | Protocatechuic | Vanillic | Rutin | Quercetin | Gallic | Catechin |
|---|---|---|---|---|---|---|---|
| *M. rubra* | 65GV01 | 0.086 ± 0.002 [e,f,*] | 0.096 ± 0.001 [d] | 2.557 ± 0.011 [d] | 0.136 ± 0.004 [f,g] | 0.154 ± 0.001 [b] | 0.059 ± 0.001 [c] |
| | 65GV02 | 0.080 ± 0.001 [g,h] | 0.089 ± 0.002 [e] | 0.144 ± 0.003 [h,i] | 0.146 ± 0.001 [d,e] | 0.163 ± 0.002 [b] | 0.015 ± 0.000 [i] |
| | 65GV03 | 0.090 ± 0.002 [c,d] | 0.091 ± 0.001 [e] | 0.150 ± 0.001 [h] | 0.142 ± 0.001 [e] | 0.096 ± 0.001 [c] | 0.032 ± 0.001 [e] |
| | 65GV04 | 0.087 ± 0.001 [d,e] | 0.091 ± 0.001 [e] | 0.190 ± 0.003 [g] | 0.131 ± 0.002 [g] | 0.047 ± 0.002 [d,e] | 0.020 ± 0.002 [h] |
| | 65GV05 | 0.086 ± 0.001 [e,f] | 0.089 ± 0.001 [e] | 0.135 ± 0.004 [i,j] | 0.149 ± 0.002 [d] | 0.084 ± 0.069 [c] | 0.011 ± 0.000 [j] |
| *M. alba* | 65GV06 | 0.104 ± 0.005 [a] | 0.091 ± 0.003 [e] | 0.380 ± 0.001 [e] | 0.176 ± 0.005 [c] | 0.019 ± 0.001 [e] | 0.026 ± 0.001 [f] |
| | 65GV07 | 0.096 ± 0.001 [b] | 0.090 ± 0.001 [e] | 0.216 ± 0.004 [f] | 0.141 ± 0.005 [e,f] | 0.151 ± 0.001 [b] | 0.014 ± 0.000 [i] |
| | 65GV08 | 0.094 ± 0.003 [b,c] | 0.090 ± 0.002 [e] | 0.158 ± 0.008 [h] | 0.143 ± 0.002 [d,e] | 0.063 ± 0.003 [c,d] | 0.090 ± 0.000 [a] |
| | 65GV09 | 0.078 ± 0.001 [h] | 0.090 ± 0.001 [e] | 0.128 ± 0.002 [j] | 0.143 ± 0.003 [e] | 0.303 ± 0.001 [a] | 0.021 ± 0.000 [h] |
| | 65GV10 | 0.097 ± 0.002 [b] | 0.108 ± 0.001 [b] | 3.882 ± 0.017 [b] | 0.225 ± 0.005 [b] | 0.153 ± 0.003 [b] | 0.078 ± 0.002 [b] |
| *M. nigra* | 65GV11 | 0.083 ± 0.002 [f,g] | 0.115 ± 0.002 [a] | 3.903 ± 0.011 [a] | 0.297 ± 0.007 [a] | 0.163 ± 0.001 [b] | 0.054 ± 0.003 [d] |
| | 65GV12 | 0.090 ± 0.002 [c,d] | 0.101 ± 0.002 [c] | 2.763 ± 0.014 [c] | 0.176 ± 0.002 [c] | 0.153 ± 0.001 [b] | 0.023 ± 0.000 [g] |

* The difference among the means indicated with the same lower-case letter in columns was not significant ($p \leq 0.05$).

**Table 8.** Chlorogenic, caffeic, syringic, *p*-coumaric, ferulic, and phlorizin content of mulberry landraces in the Gevaş District of Van (mg g$^{-1}$).

| Species | Landraces | Chlorogenic | Caffeic | Syringic | *p*-Coumaric | Ferulic | Phlorizin |
|---|---|---|---|---|---|---|---|
| *M. rubra* | 65GV01 | 1.632 ± 0.003 [c,*] | 0.121 ± 0.001 [d,e] | 0.118 ± 0.003 [g] | 0.081 ± 0.001 [b] | 0.093 ± 0.003 [b] | 0.042 ± 0.001 [e] |
| | 65GV02 | 1.360 ± 0.005 [f] | 0.116 ± 0.003 [f,g] | 0.138 ± 0.001 [a] | 0.081 ± 0.004 [b] | 0.094 ± 0.005 [b] | 0.033 ± 0.001 [f] |
| | 65GV03 | 1.329 ± 0.011 [g] | 0.130 ± 0.005 [c] | 0.142 ± 0.001 [a] | 0.077 ± 0.000 [b] | 0.095 ± 0.006 [b] | 0.081 ± 0.001 [b] |
| | 65GV04 | 1.247 ± 0.036 [i] | 0.124 ± 0.002 [d] | 0.122 ± 0.002 [e] | 0.083 ± 0.001 [b] | 0.524 ± 0.426 [a] | 0.019 ± 0.000 [h] |
| | 65GV05 | 1.366 ± 0.003 [f] | 0.122 ± 0.003 [d] | 0.129 ± 0.004 [c] | 0.425 ± 0.345 [a] | 0.090 ± 0.001 [b] | 0.016 ± 0.001 [i] |
| *M. alba* | 65GV06 | 1.411 ± 0.006 [e] | 0.136 ± 0.005 [b] | 0.127 ± 0.003 [d] | 0.091 ± 0.001 [b] | 0.095 ± 0.005 [b] | 0.025 ± 0.002 [g] |
| | 65GV07 | 1.332 ± 0.001 [g] | 0.112 ± 0.000 [g,h] | 0.133 ± 0.002 [b] | 0.087 ± 0.001 [b] | 0.092 ± 0.002 [b] | 0.013 ± 0.002 [j] |
| | 65GV08 | 1.486 ± 0.012 [d] | 0.109 ± 0.001 [h] | 0.124 ± 0.004 [d,e] | 0.082 ± 0.002 [b] | 0.091 ± 0.001 [b] | 0.017 ± 0.000 [i] |
| | 65GV09 | 1.286 ± 0.013 [h] | 0.117 ± 0.001 [e,f] | 0.133 ± 0.005 [b] | 0.079 ± 0.003 [b] | 0.094 ± 0.004 [b] | 0.014 ± 0.001 [j] |
| | 65GV10 | 0.988 ± 0.004 [j] | 0.134 ± 0.001 [b,c] | 0.120 ± 0.002 [f,g] | 0.123 ± 0.002 [b] | 0.092 ± 0.003 [b] | 0.166 ± 0.001 [a] |
| *M. nigra* | 65GV11 | 2.314 ± 0.006 [b] | 0.137 ± 0.001 [b] | 0.131 ± 0.001 [b,c] | 0.091 ± 0.003 [b] | 0.095 ± 0.000 [b] | 0.073 ± 0.001 [c] |
| | 65GV12 | 3.526 ± 0.003 [a] | 0.160 ± 0.002 [a] | 0.131 ± 0.004 [b,c] | 0.079 ± 0.002 [b] | 0.093 ± 0.002 [b] | 0.058 ± 0.001 [d] |

* The difference among the means indicated with the same lower-case letter in columns was not significant ($p \leq 0.05$).

**Table 9.** Protocatechuic, vanillic, rutin, quercetin, gallic, and catechin content of mulberry landraces in the Adilcevaz District of Bitlis (mg g$^{-1}$).

| Species | Landraces | Protocatechuic | Vanillic | Rutin | Quercetin | Gallic | Catechin |
|---|---|---|---|---|---|---|---|
| | 13AD01 | 0.023 ± 0.003 j,* | 0.078 ± 0.002 h | 1.914 ± 0.011 d | 0.055 ± 0.003 e | 0.153 ± 0.002 b,c,d | 0.070 ± 0.002 i |
| | 13AD02 | 0.016 ± 0.001 k,l | 0.016 ± 0.001 j | 2.523 ± 0.022 a | 0.044 ± 0.002 e,f | 0.158 ± 0.003 b,c | 0.146 ± 0.003 c |
| | 13AD03 | 0.011 ± 0.001 l | 0.033 ± 0.001 i,j | 1.047 ± 0.047 g,h | 0.038 ± 0.002 e,f | 0.155 ± 0.001 b,c,d | 0.051 ± 0.001 k |
| | 13AD04 | 0.022 ± 0.001 j | 0.065 ± 0.002 h,i | 0.882 ± 0.009 i | 0.012 ± 0.000 f | 0.158 ± 0.004 b,c | 0.052 ± 0.001 j,k |
| | 13AD05 | 0.055 ± 0.003 g | 0.286 ± 0.007 a | 1.459 ± 0.016 e | 0.063 ± 0.001 e | 0.154 ± 0.000 b,c,d | 0.103 ± 0.001 e,f |
| *M. rubra* | 13AD06 | 0.029 ± 0.013 i | 0.232 ± 0.100 b,c | 0.907 ± 0.392 h,i | 0.185 ± 0.080 a,b | 0.273 ± 0.118 a | 0.063 ± 0.027 l,j |
| | 13AD07 | 0.083 ± 0.001 f | 0.193 ± 0.002 c,d | 0.928 ± 0.025 h,i | 0.162 ± 0.002 b,c,d | 0.156 ± 0.001 b,c,d | 0.050 ± 0.001 k |
| | 13AD08 | 0.084 ± 0.003 f | 0.215 ± 0.013 c,d | 1.997 ± 0.039 c,d | 0.137 ± 0.003 d | 0.146 ± 0.003 b,c,d | 0.158 ± 0.001 b |
| | 13AD09 | 0.103 ± 0.003 c | 0.123 ± 0.002 f,g | 1.188 ± 0.009 f,g | 0.144 ± 0.005 c,d | 0.156 ± 0.003 b,c,d | 0.124 ± 0.000 d |
| | 13AD10 | 0.085 ± 0.001 e,f | 0.094 ± 0.002 g,h | 1.126 ± 0.013 f,g | 0.136 ± 0.004 d | 0.107 ± 0.003 c,d,e | 0.096 ± 0.000 f,g |
| | 13AD11 | 0.107 ± 0.002 c | 0.088 ± 0.003 g,h | 2.129 ± 0.027 c | 0.150 ± 0.008 c,d | 0.154 ± 0.004 b,c,d | 0.152 ± 0.001 b,c |
| | 13AD12 | 0.226 ± 0.006 a | 0.173 ± 0.002 d,e | 0.182 ± 0.004 j | 0.177 ± 0.001 a,b,c | 0.097 ± 0.002 d,e | 0.084 ± 0.001 h |
| | 13AD13 | 0.189 ± 0.001 b | 0.175 ± 0.003 d,e | 1.461 ± 0.016 e | 0.148 ± 0.002 c,d | 0.157 ± 0.001 b,c | 0.131 ± 0.003 d |
| | 13AD14 | 0.090 ± 0.000 e | 0.087 ± 0.003 g,h | 0.122 ± 0.001 j | 0.143 ± 0.003 c,d | 0.252 ± 0.004 a | 0.027 ± 0.002 l |
| *M. alba* | 13AD15 | 0.097 ± 0.001 d | 0.091 ± 0.001 g,h | 0.176 ± 0.000 j | 0.150 ± 0.003 c,d | 0.163 ± 0.004 b,c | 0.065 ± 0.002 i |
| | 13AD16 | 0.060 ± 0.002 g | 0.064 ± 0.004 h,i | 1.509 ± 0.006 e | 0.062 ± 0.002 e | 0.160 ± 0.003 b,c | 0.123 ± 0.001 d |
| | 13AD17 | 0.040 ± 0.000 h | 0.213 ± 0.003 c,d | 1.252 ± 0.015 f | 0.019 ± 0.000 f | 0.084 ± 0.069 e | 0.068 ± 0.001 i |
| *M. nigra* | 13AD18 | 0.018 ± 0.000 j,k | 0.265 ± 0.005 a,b | 2.531 ± 0.016 a | 0.206 ± 0.004 a | 0.163 ± 0.001 b,c | 0.107 ± 0.003 e |
| | 13AD19 | 0.036 ± 0.001 h | 0.104 ± 0.004 g,h | 2.374 ± 0.000 b | 0.161 ± 0.007 b,c,d | 0.169 ± 0.002 b | 0.183 ± 0.002 a |
| | 13AD20 | 0.089 ± 0.002 ef | 0.150 ± 0.005 ef | 1.172 ± 0.011 fg | 0.137 ± 0.002 d | 0.153 ± 0.001 b,c,d | 0.092 ± 0.002 g,h |

\* The difference among the means indicated with the same lower-case letter in columns was not significant ($p \leq 0.05$).

**Table 10.** Chlorogenic, caffeic, syringic, *p*-coumaric, ferulic, and phlorizin content of mulberry landraces in the Adilcevaz District of Bitlis (mg g$^{-1}$).

| Species | Landraces | Chlorogenic | Caffeic | Syringic | *p*-Coumaric | Ferulic | Phlorizin |
|---|---|---|---|---|---|---|---|
| | 13AD01 | 1.560 ± 0.002 c,d,* | 0.183 ± 0.003 e,f | 0.118 ± 0.002 c,d | 0.084 ± 0.004 c,d,e | 0.101 ± 0.001 b | 0.057 ± 0.002 f,g |
| | 13AD02 | 0.547 ± 0.003 h,i | 0.087 ± 0.001 h | 0.094 ± 0.003 d,e | 0.082 ± 0.001 e | 0.110 ± 0.000 a,b | 0.042 ± 0.001 f,g |
| | 13AD03 | 0.931 ± 0.003 f,g | 0.213 ± 0.005 d,e | 0.115 ± 0.001 c,d | 0.088 ± 0.006 c,d,e | 0.095 ± 0.003 b | 0.070 ± 0.001 f,g |
| | 13AD04 | 1.225 ± 0.014 e | 0.385 ± 0.005 a | 0.075 ± 0.003 e | 0.089 ± 0.003 c,d,e | 0.094 ± 0.006 b | 0.069 ± 0.003 f,g |
| | 13AD05 | 1.949 ± 0.005 b | 0.310 ± 0.005 b,c | 0.119 ± 0.002 c,d | 0.107 ± 0.005 a,b,c,d | 0.091 ± 0.000 b | 0.032 ± 0.002 g |
| *M. rubra* | 13AD06 | 1.333 ± 0.577 d,e | 0.300 ± 0.130 b,c | 0.202 ± 0.088 a | 0.120 ± 0.052 a | 0.128 ± 0.055 a | 0.356 ± 0.154 a |
| | 13AD07 | 1.453 ± 0.046 c,d | 0.263 ± 0.011 c,d | 0.125 ± 0.001 c,d | 0.095 ± 0.004 b,c,d,e | 0.094 ± 0.005 b | 0.207 ± 0.07 b,c,d |
| | 13AD08 | 2.461 ± 0.025 a | 0.352 ± 0.007 a,b | 0.118 ± 0.001 c,d | 0.107 ± 0.001 a,b,c | 0.089 ± 0.001 b | 0.239 ± 0.033 b,c |
| | 13AD09 | 0.745 ± 0.015 g,h | 0.187 ± 0.001 e,f | 0.144 ± 0.005 b,c | 0.090 ± 0.001 c,d,e | 0.091 ± 0.002 b | 0.206 ± 0.001 b,c,d |
| | 13AD10 | 0.328 ± 0.009 i,j | 0.113 ± 0.002 g,h | 0.130 ± 0.002 b,c,d | 0.096 ± 0.002 b,c,d,e | 0.095 ± 0.002 b | 0.253 ± 0.004 b |
| | 13AD11 | 0.111 ± 0.004 j | 0.092 ± 0.001 h | 0.123 ± 0.006 c,d | 0.077 ± 0.001 e | 0.094 ± 0.004 b | 0.160 ± 0.005 d,e |
| | 13AD12 | 1.164 ± 0.002 e,f | 0.216 ± 0.004 d,e | 0.122 ± 0.001 c,d | 0.087 ± 0.003 c,d,e | 0.095 ± 0.004 b | 0.185 ± 0.002 c,d |
| | 13AD13 | 0.966 ± 0.041 f,g | 0.202 ± 0.005 e,f | 0.122 ± 0.002 c,d | 0.094 ± 0.003 c,d,e | 0.091 ± 0.001 b | 0.178 ± 0.004 c,d |
| | 13AD14 | 0.106 ± 0.002 j | 0.089 ± 0.001 h | 0.120 ± 0.001 c,d | 0.076 ± 0.002 e | 0.099 ± 0.001 b | 0.108 ± 0.001 e,f |
| *M. alba* | 13AD15 | 0.110 ± 0.008 j | 0.094 ± 0.001 h | 0.127 ± 0.001 b,c,d | 0.079 ± 0.000 e | 0.097 ± 0.002 b | 0.151 ± 0.001 d,e |
| | 13AD16 | 0.545 ± 0.004 h,i | 0.268 ± 0.002 c | 0.119 ± 0.001 c,d | 0.024 ± 0.000 f | 0.094 ± 0.002 b | 0.085 ± 0.005 f,g |
| | 13AD17 | 1.526 ± 0.014 c,d | 0.268 ± 0.002 c | 0.021 ± 0.000 f | 0.083 ± 0.001 e | 0.093 ± 0.002 b | 0.073 ± 0.002 f,g |
| *M. nigra* | 13AD18 | 1.637 ± 0.009 c | 0.351 ± 0.005 a,b | 0.162 ± 0.003 b | 0.117 ± 0.002 a,b | 0.090 ± 0.002 b | 0.175 ± 0.002 c,d |
| | 13AD19 | 0.556 ± 0.004 h,i | 0.151 ± 0.001 f,g | 0.120 ± 0.002 c,d | 0.084 ± 0.004 d,e | 0.091 ± 0.001 b | 0.079 ± 0.000 f,g |
| | 13AD20 | 0.994 ± 0.004 f | 0.199 ± 0.005 ef | 0.127 ± 0.003 b,c,d | 0.097 ± 0.001 b,c,d,e | 0.087 ± 0.004 b | 0.040 ± 0.001 f,g |

\* The difference among the means indicated with the same lower-case letter in columns was not significant ($p \leq 0.05$).

**Table 11.** Protocatechuic, vanillic, rutin, quercetin, gallic, and catechin content of mulberry landraces in the Ahlat District of Bitlis (mg g$^{-1}$).

| Species | Landraces | Protocatechuic | Vanillic | Rutin | Quercetin | Gallic | Catechin |
|---|---|---|---|---|---|---|---|
| | 13AH01 | 0.103 ± 0.003 c,* | 0.145 ± 0.005 c | 1.365 ± 0.014 c | 0.095 ± 0.005 h | 0.147 ± 0.008 h | 0.102 ± 0.004 h |
| | 13AH02 | 0.083 ± 0.003 e,f | 0.601 ± 0.001 a | 1.705 ± 0.014 b | 0.125 ± 0.005 g | 0.159 ± 0.001 g | 0.123 ± 0.001 g |
| *M. rubra* | 13AH03 | 0.136 ± 0.007 a | 0.205 ± 0.005 b | 0.875 ± 0.020 d | 0.182 ± 0.004 b | 0.155 ± 0.005 gh | 0.164 ± 0.001 d |
| | 13AH04 | 0.136 ± 0.004 a | 0.145 ± 0.004 c | 2.229 ± 0.077 a | 0.144 ± 0.003 e | 0.147 ± 0.007 h | 0.185 ± 0.001 c |
| | 13AH05 | 0.088 ± 0.002 d,e | 0.089 ± 0.001 e | 0.531 ± 0.000 e | 0.131 ± 0.001 f,g | 0.192 ± 0.012 e | 0.025 ± 0.001 i |
| | 13AH06 | 0.117 ± 0.002 b | 0.090 ± 0.001 e | 0.144 ± 0.001 h | 0.134 ± 0.004 f | 0.147 ± 0.007 h | 0.015 ± 0.000 j |
| | 13AH07 | 0.087 ± 0.002 d,e | 0.099 ± 0.006 d | 0.177 ± 0.017 h | 0.155 ± 0.002 d | 0.175 ± 0.005 f | 0.250 ± 0.002 b |
| | 13AH08 | 0.086 ± 0.001 d,e | 0.090 ± 0.001 e | 0.240 ± 0.003 g | 0.130 ± 0.002 f,g | 0.152 ± 0.000 gh | 0.266 ± 0.003 a |
| *M. alba* | 13AH09 | 0.079 ± 0.003 f,g | 0.090 ± 0.003 e | 0.174 ± 0.006 h | 0.133 ± 0.002 f,g | 0.326 ± 0.003 a | 0.269 ± 0.002 a |
| | 13AH10 | 0.115 ± 0.002 b | 0.091 ± 0.005 e | 0.167 ± 0.008 h | 0.129 ± 0.002 f,g | 0.241 ± 0.001 d | 0.134 ± 0.004 e |
| | 13AH11 | 0.088 ± 0.003 d,e | 0.089 ± 0.000 e | 0.246 ± 0.003 g | 0.133 ± 0.003 f,g | 0.320 ± 0.002 a | 0.164 ± 0.002 d |
| | 13AH12 | 0.078 ± 0.001 g | 0.089 ± 0.001 e | 0.179 ± 0.006 h | 0.278 ± 0.012 a | 0.306 ± 0.006 b | 0.125 ± 0.001 f,g |
| | 13AH13 | 0.091 ± 0.001 d | 0.091 ± 0.001 e | 0.428 ± 0.007 f | 0.170 ± 0.001 c | 0.252 ± 0.002 c | 0.128 ± 0.004 f |

\* The difference among the means indicated with the same lower-case letter in columns was not significant ($p \leq 0.05$).

**Table 12.** Chlorogenic, caffeic, syringic, *p*-coumaric, ferulic and phlorizin content of mulberry landraces in the Ahlat District of Bitlis (mg g$^{-1}$).

| Species | Landraces | Chlorogenic | Caffeic | Syringic | *p*-Coumaric | Ferulic | Phlorizin |
|---|---|---|---|---|---|---|---|
| *M. rubra* | 13AH01 | 2.145 ± 0.015 [c,*] | 0.245 ± 0.006 [a] | 0.113 ± 0.003 [e,f] | 0.115 ± 0.005 [a] | 0.100 ± 0.002 [a] | 0.061 ± 0.002 [f] |
| | 13AH02 | 6.246 ± 0.046 [a] | 0.251 ± 0.009 [a] | 0.109 ± 0.009 [f] | 0.087 ± 0.008 [c,d,e] | 0.093 ± 0.004 [a] | 0.066 ± 0.002 [f] |
| | 13AH03 | 6.076 ± 0.592 [a] | 0.208 ± 0.008 [b] | 0.142 ± 0.011 [a] | 0.094 ± 0.004 [b,c] | 0.095 ± 0.005 [a] | 0.200 ± 0.007 [b] |
| | 13AH04 | 5.068 ± 0.797 [b] | 0.171 ± 0.007 [c] | 0.119 ± 0.001 [d,e] | 0.094 ± 0.001 [b,c] | 0.045 ± 0.045 [b] | 0.246 ± 0.008 [a] |
| | 13AH05 | 1.340 ± 0.038 [d] | 0.089 ± 0.003 [f] | 0.125 ± 0.006 [c,d] | 0.082 ± 0.001 [d,e] | 0.091 ± 0.001 [a] | 0.199 ± 0.005 [b] |
| | 13AH06 | 1.331 ± 0.057 [d] | 0.092 ± 0.005 [e,f] | 0.131 ± 0.005 [b,c] | 0.091 ± 0.009 [b,c,d] | 0.094 ± 0.003 [a] | 0.121 ± 0.009 [e] |
| | 13AH07 | 1.421 ± 0.049 [d] | 0.096 ± 0.001 [d,e,f] | 0.138 ± 0.002 [a,b] | 0.077 ± 0.008 [e] | 0.103 ± 0.011 [a] | 0.128 ± 0.001 [de] |
| *M. alba* | 13AH08 | 1.314 ± 0.062 [d] | 0.088 ± 0.001 [f,g] | 0.124 ± 0.006 [c,d] | 0.081 ± 0.004 [d,e] | 0.096 ± 0.002 [a] | 0.144 ± 0.005 [c] |
| | 13AH09 | 1.299 ± 0.045 [d] | 0.105 ± 0.006 [d] | 0.128 ± 0.005 [c,d] | 0.085 ± 0.006 [c,d,e] | 0.096 ± 0.002 [a] | 0.121 ± 0.001 [e] |
| | 13AH10 | 1.333 ± 0.011 [d] | 0.092 ± 0.008 [e,f] | 0.125 ± 0.002 [c,d] | 0.094 ± 0.004 [b,c] | 0.091 ± 0.001 [a] | 0.129 ± 0.005 [de] |
| | 13AH11 | 1.235 ± 0.003 [d] | 0.080 ± 0.000 [g] | 0.130 ± 0.000 [b,c] | 0.090 ± 0.001 [c,d] | 0.092 ± 0.001 [a] | 0.135 ± 0.007 [d] |
| | 13AH12 | 1.414 ± 0.058 [d] | 0.090 ± 0.001 [f] | 0.130 ± 0.006 [b,c] | 0.090 ± 0.001 [c,d] | 0.090 ± 0.001 [a] | 0.131 ± 0.003 [d] |
| | 13AH13 | 1.287 ± 0.025 [d] | 0.100 ± 0.001 [d,e] | 0.120 ± 0.001 [d,e] | 0.101 ± 0.009 [b] | 0.092 ± 0.001 [a] | 0.200 ± 0.000 [b] |

\* The difference among the means indicated with the same lower-case letter in columns was not significant ($p \leq 0.05$).

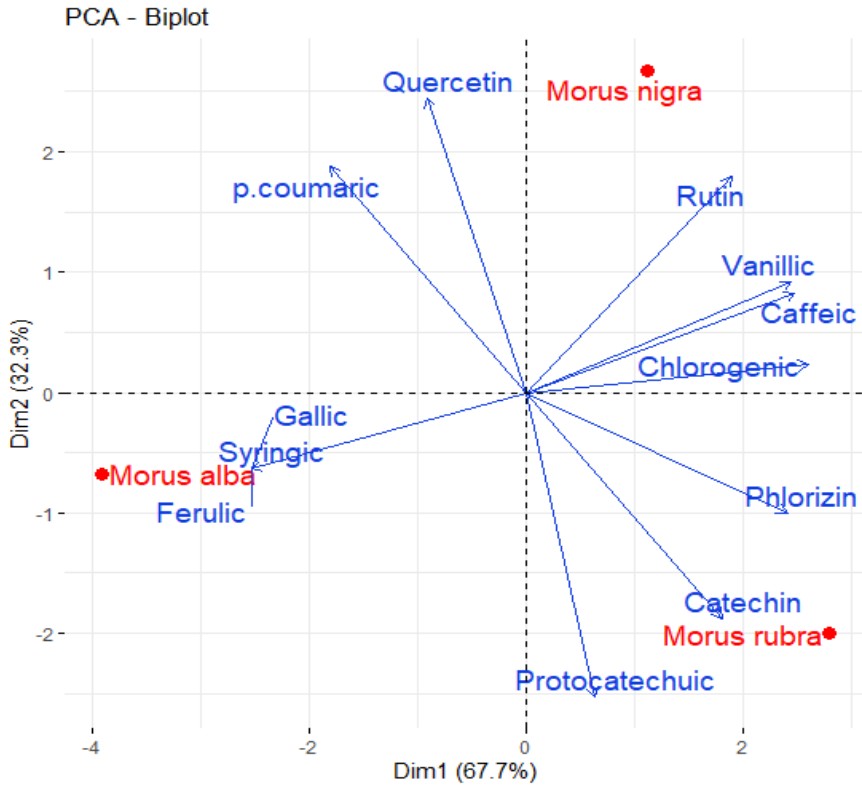

**Figure 4.** Correlation between mulberry species in terms of phenolic compounds.

## 4. Discussion

Organic acids are of great importance in human health and are effective in many physiological events (taste formation, ripening, etc.) in plants. The organic acid sugar ratio reveals the ripening status of the fruits [12,21]. When these acids are low, the fruits are sweet, and when they are high, the fruits become sour [37]. Özgen et al. [38] and Sanchez et al. [25] reported that malic acid and citric acid were higher than other organic acids in mulberry fruit. Gundogdu et al. [21] reported that citric acid, tartaric acid, malic acid, succinic acid, and fumaric acid in black mulberries were 1.084 g 100 g$^{-1}$, 0.123 g 100 g$^{-1}$, 1.323 g 100 g$^{-1}$, 0.342 g 100 g$^{-1}$, and 0.011 g 100 g$^{-1}$, respectively. Eyduran et al. [39] reported that the malic acid content of mulberry fruits ranged from 1.13 to 3.04 g 100 g$^{-1}$. Geçer et al. [40] found that the malic acid, citric acid, tartaric acid, succinic acid, and fumaric acid contents of mulberries were 3.07–2.13 g 100 g$^{-1}$, 0.48–1.03 g 100 g$^{-1}$, 0.15–0.43 g 100 g$^{-1}$, 0.12–0.44 g 100 g$^{-1}$, 0.01–0.12 g 100 g$^{-1}$, respectively. Ercisli and Orhan [22] determined that the vitamin C content of fruits from black

mulberry genotypes grown in the Northeastern Anatolia region of Turkey ranged from 14.9 to 18.8 mg 100 mL$^{-1}$. It has been observed that the results we have obtained in this study are generally in parallel with the findings of other researchers [25,39–41]. The vitamin C content of fruits varies according to the different extraction methods. Hadeel et al. [42] found that the vitamin C content of bananas was lower when extracted with methanol. The same researchers revealed that the vitamin C content was higher in pomegranates and white grapes. Biochemical compounds vary according to the different parts of the fruit and applications affect these compounds [43–46]. In addition, cultural practices and genetic and environmental factors affect the examined genotypes' organic acid and vitamin C contents. Although the losses of organic acids and vitamin C were minimized during fruit harvest, storage, and analysis processes, this situation could not be prevented entirely. Therefore, changes and reactions in the physiology of fruits affected the organic acid and vitamin C content.

Phenolic compounds react with the enzyme polyphenoloxidase and cause browning. In this respect, these compounds are biochemicals that are effective in fruit processing and color formation. It is influential in forming the flavors of the products, especially in forming a bitter taste in the mouth [12,21]. Anthocyanins, one of the phenolic substances, provide the unique colors of fruits and vegetables. In addition, phenolic compounds cause the browning of products obtained from fruits and vegetables. Phenolic compounds, which significantly affect the fruit juice processing industry, form deposits in beverages such as fruit juices and wine [34]. According to Gundogdu et al. [21] and Natic et al. [24], chlorogenic acid and rutin were the two main phenolics in mulberry fruits. Considering the phenolic substance distributions of the examined genotypes, significant differences were determined between them. It is thought that these differences are caused by species, genotype characteristics, cultural practices (fertilization, pruning etc.,), climate and soil characteristics of the region where they are grown. According to Gundogdu et al. [21], the amounts of gallic acid, catechin, chlorogenic acid, caffeic acid, syringic acid, *p*-coumaric acid, ferulic acid, o-coumaric acid, vanillic acid, rutin, and quercetin in black mulberry fruits were 0.15 mg g$^{-1}$, 0.08 mg g$^{-1}$, 3.11 mg g$^{-1}$, 0.13 mg g$^{-1}$, 0.10 mg g$^{-1}$, 0.13 mg g$^{-1}$, 0.06 mg g$^{-1}$, 0.13 mg g$^{-1}$, 0.04 mg g$^{-1}$, 1.42 mg g$^{-1}$, 0.11 mg g$^{-1}$, respectively. Eyduran et al. [39] reported that rutin and chlorogenic acid contents were higher than other phenolic compounds in mulberry fruits. Gecer et al. [40] reported that routine in black mulberry fruits and chlorogenic acid in white mulberries was high. Among the factors affecting the phenolic compounds in fruits are genetic factors, ecological factors (such as humidity, light, temperature, and soil structure), and cultural practices [21,38,40,41,47–51]. In addition, phenolic substances affect the products obtained by processing fruits to leave a bitter taste in the mouth after consumption and have a cloudy fruit juices' appearance [30]. Therefore, especially phenolic compounds play an important role in the fruit juice processing industry.

## 5. Conclusions

In this study, it was seen that the Van Lake basin has rich mulberry genetic resources. It had been observed that some landraces stand out in terms of biochemical contents among these mulberry genetic resources. These landraces are evaluated on a basin basis; 65VN09 landrace in the Gevaş region, 65VN09 genotype in Van Merkez, 13AD16 genotype in the Adilcevaz region and 13AH12 landrace in Ahlat region were found promising in terms of malic acid content, which is the dominant organic acid. When landraces are evaluated in terms of chlorogenic acid content, which is the dominant phenolic among phenolic compounds, it was seen that 65VN03 genotype in the Van Central region, 65GV12 genotype in Gevaş region, 13AD08 landrace in Adilcevaz region, and 13AH02 landrace in Ahlat region came to the fore.

**Author Contributions:** Conceptualization and methodology, A.C., A.K. and E.O.; data curation, A.C., A.K. and E.O.; investigation, M.G., A.C. and A.K.; writing—original draft, S.E., R.K. and R.C.; writing—review and editing, S.E., R.K. and R.C. All authors have read and agreed to the published version of the manuscript.

**Funding:** This research received no external funding.

**Institutional Review Board Statement:** Not applicable.

**Informed Consent Statement:** Not applicable.

**Data Availability Statement:** All-new research data were presented in this contribution.

**Conflicts of Interest:** The authors declare that they have no conflict of interest.

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
