# Peer review of "Sustainable Mulberry (Morus nigra L., Morus alba L. and Morus rubra L.) Production in Eastern Turkey"

_sustainability, doi:10.3390/su132413507_

Round 1
Reviewer 1 Report
To the Authors (in detail)
- 1 sub-section: from how many plants were collected fruits?
- 1 section, line 223 and in the whole manuscript, when you indicate the temperature, separate the numeric value from the symbol: -80 °C and not -80°C;
- 3 sub-section, line 154, verify how to write the particle size: 5 and not five;
- 3 section, line 302, verify the black color of 1 and min-1;
- Section 5, line 582, please, verify, it is not comprehensible… fig…;
- References
- References 6, 24-25 and in the whole section, verify the punctuation and the spacing between words and between words and punctuation:
- Please, apply carefully all the comments I have suggested, include a point by point list of your corrections and write in blue color, in the text, the corrections you will do.
Regards.
Reviewer 2 Report
The material has been improved in accordance with the suggestions submitted previously.
This manuscript is a resubmission of an earlier submission. The following is a list of the peer review reports and author responses from that submission.
Round 1
Reviewer 1 Report
This manuscript deals with metabolic analysis of mulberry fruits collected from 55 trees in specific region of Van lake in eastern Turkey.
In my view the analysis is conducted properly, however it represents only single year analysis and can not capture the variation.
Furthemore I think the study is more suited for other food analysis related journal such as Foods (MDPI) rather than sustainability. The only argument for sustainability is given in the title.. based on pestide free production on farmers fields.
This is truly solely chemical analysis which has nothing to do with sustainability.
specific points:
I do not agree that these are wild trees - since this is already domesticated species. What you want to say is that these trees are not specifically breed and neither represent specific variety, but rather landraces.
Statement that there is 100% variation (lines 28-29 of Abstract) is incorrect and missleading.
How the material was stored until analysis ? was frozen or freeze dried?
Reviewer 2 Report
Overall, the topic proposed by the manuscript is interesting for the readers and in line with the journal`s aims and scope.
The article titled "Sustainable mulberry (Morus nigra L., Morus alba L. and Morus rubra L.) production in eastern Turkey" presents an interesting approach in terms of analyzing the qualitative and compositional differences for three varieties of mulberry, Morus nigra L ., Morus alba L., and Morus rubra L, found around the Van Lake basin, Turkey.
The authors present analytically the chemical characteristics of a significant number of genotypes, a total of 55 wild grown mulberry genotypes.
The introduction is done correctly, presenting the specifics of the area of ​​origin of the analyzed material as well as the particularities of mulberry cultivation in Turkey.
The methodology used by the authors is specific to chemical determinations for plant products and is correctly applied.
Regarding the analysis and interpretation of the obtained data, the authors used appropriate methods and software to obtain significant results.
However, the paper has several aspects that could be improved, namely:
1. Chapter Results and discussion should be separated into two distinct chapters
2. The conclusions of the study are partially supported by the results of the study. For example, there is no justification for the statement This type of fruit, which is widely used, cannot meet the market demand since its cultivation and production is limited. I consider it risky as long as there is no clear analysis in the study regarding the demand and supply for this product category.
3. Authors must present the limitations of the study.
Reviewer 3 Report
Manuscript Number: sustainability-1410646, titled:
Sustainable mulberry (Morus nigra L., Morus alba L. and Morus rubra L.) production in eastern Turkey
Review 1 – 7 October 2021
To the Authors (in detail)
- To avoid confusion, please use the correct and updated botanical nomenclature, for example according to www.gbif.org, in the title and at the first mention in the text, after genus, specie and botanist, insert the botanical family;
- 1 sub-section, please, include the description of the climatic conditions, type of soil,;
- 1 sub-section: from how many plants were collected fruits?
- 1 sub-section: describe the age of plants;
- 1 sub-section, lines 123-124,127 and in the whole manuscript: the scientific names in italic font;
- 2 sub-section: line 146: 1 ml min-1: use the superscript for -1 or use the fraction;
- 2 sub-section, line 146: replace ml with mL;
- 3 sub-section, lines 150 and 152, please verify if % is before or after 2.5;
- 3 sub-section, line 151: A 0.5 aliquot of the mixture;
- 3 sub-section: lines 150, 152 and in the whole manuscript, w v-1: use the superscript for -1 or use the fraction;
- 3 sub-section, line 154, verify how to write the particle size;
- 3 sub-section, line 158 and in the whole manuscript: replace ppm with mg/L or mg/kg;
- Caption of table 1, mg/100 g: in your manuscript you have used sometime the fraction and sometime the exponent, please, be consistent and use always the same system in the whole manuscript, tables and figures;
- Caption of table 2 and in the whole manuscript, when you indicate a quantity, separate the numeric value from the symbol: 100 g and not 100g;
- 1 section, compare your data with findings of other authors in different geographical areas of the world;
- 1 sub-section, vitamin C is a very important parameter and it is contained in many fruits and vegetables but the discussion is not well treated in your manuscript. Compare your data with findings of other authors in different fruits and vegetables. Here find, read and discuss:
[X1] Effect of different extraction solvents on the antioxidant content and capacity of nine seasonal fruits.
Clinical Nutrition Open Science 38, 33 – 42 (2021). DOI
10.1016/j.nutos.2021.06.003
[X2] Citrus bergamia, Risso: the peel, the juice and the seed oil of the bergamot fruit of Reggio Calabria (South Italy).
Emirates Journal of Food and Agriculture 32(7) 522-532 (2020).
DOI: 10.9755/ejfa.2020.v32.i7.2128
[X3] Physicochemical, nutritional, and bioactive properties of pulp and peel from 15 kiwifruit cultivars.
Food Bioscience 42 (2021) 101157. DOI: 10.1016/j.fbio.2021.101157
[X4] Use of digestate as an alternative to mineral fertilizer: effects on growth and crop quality.
Archives of Agronomy and Soil Science 65 (5) 700-711 (2019).
https://doi.org/10.1080/03650340.2018.1520980
[X5] The combination of selenium and LED light quality affects growth and nutritional properties of broccoli sprouts.
Molecules 25 (20) 2020 Article number 4788. DOI: 10.3390/molecules25204788
- 1 sub-section, in your discussion explain if the differences between the geographical areas are due to the microclimatic conditions, because the genotype cannot explain the 100% of chemical diversity of fruits;
- 1 sub-section, lines 227-229, support your statement with some reference;
- 1 sub-section, when you discuss about results of other authors, describe when the fruits of these studies were sampled, in addition include some information more such as the origin of fruits and if they are from wild or cultivated plants;
- 2 sub-section, line 251: routine content? Please, verify;
- 2 sub-section, lines 311-312, extend this statement, discuss better and support it with some other reference in relation with color and anthocyanins:
- References section, ref 21, in the title of the paper, separate the genus from the specie;
- References section, please verify carefully the whole section, in particular verify if to insert one space before the publication year;
- Please, apply carefully all the comments I have suggested, include a point by point list of your corrections and evidence in the text the corrections you will do.
Regards.
